# The Influence of Adherence to Orthosis and Physiotherapy Protocol on Functional Outcome after Proximal Humeral Fracture in the Elderly

**DOI:** 10.3390/jcm12051762

**Published:** 2023-02-22

**Authors:** Evi Fleischhacker, Johannes Gleich, Vera Smolka, Carl Neuerburg, Wolfgang Böcker, Tobias Helfen

**Affiliations:** Department of Orthopedics and Trauma Surgery, Musculoskeletal University Center Munich (MUM), University Hospital, Ludwig-Maximilians University Munich, 81377 Munich, Germany

**Keywords:** proximal humeral fracture, elderly, rehabilitation

## Abstract

In the treatment of proximal humeral fractures (PHF), patients are often recommended to wear a sling or orthosis and to perform physiotherapy. However, some patients, especially elderly ones, struggle to comply with these rehabilitation regimens. Therefore, the aim of the study was to evaluate whether these incompliant patients have a worse functional outcome than those who adhere to the rehabilitation protocol. After receiving a diagnosis of a PHF, patients were divided into four groups according to fracture morphology: conservative with sling, operative with sling, conservative with abduction orthosis, and operative with abduction orthosis. At the 6-week follow-up, compliance regarding brace use and physiotherapy performance were assessed, as well as the constant score (CS) and complications or revision surgeries. The CS as well as the complications and revision surgeries were also surveyed after one year. In 149 participants, with a mean age of 73.9 ± 7.2 years, only 37% terminated orthosis and only 49% underwent physiotherapy as recommended. The statistical analysis showed no significant difference in the CS, complications, and revision surgeries between the groups.

## 1. Introduction

Proximal humeral fracture is the third most common fracture among the elderly and it is associated with osteoporosis [1]. With an increasing life expectancy and with respect to demographic change, a remarkable increase in these fractures is expected in coming decades [2]. While non-displaced fractures may be treated non-operatively with good functional results, displaced and comminuted fractures are an indication for surgery [3,4]. Locking plates have been established for fixation after an open reduction in proximal humerus fractures over recent years [5,6]. However, due to a reduced bone quality in elderly patients, complications are observed more frequently, thus primary arthroplasty is recommended for some fracture types [7]. Regardless of the treatment (conservative vs. operative) or fracture type, physiotherapy and sometimes immobilisation are usually recommended during the first six weeks [8,9,10]. Various orthoses are available that differ in the position in which they immobilise the arm. Simple arm slings fix the arm to the body so that it is immobilised in an internal rotation and slight adduction [11]. Others achieve a gentle abduction and external rotation by placing a cushion between the arm and the body. The idea behind this is to neutralise the traction forces exerted by the supraspinatus muscle on the greater tuberosity and thus prevent cranialisation of the tuberosity [12,13]. Assuming that fractures of the greater tuberosity can be seen as bony lesions of the supraspinatus tendon, abduction orthoses are also recommended for these fracture types.

In particular, for the shoulder with its multiple injuries, there are differentiated, evidence-based and elaborated treatment protocols [8,14,15,16,17]. A 2021 paper examined the different rehabilitation protocols in the National Health Service (NHS) for proximal humerus fractures and found different recommendations for choice of orthosis, duration of immobilisation, and onset of mobilisation [18]. A review by Schnackers et al. listed five studies that used different physical therapy protocols and differed primarily in explicit arrangement into phases [19].

A certain degree of compliance seems to be necessary for the success of any therapy, not only in orthopaedics and trauma surgery [20,21]. However, wearing orthoses as well as a controlled physiotherapeutic follow-up are particularly challenging for mostly elderly patients [22,23,24]. The COVID-19 pandemic has further exacerbated the situation with regard to regular physical therapy treatment because due to increased hygiene requirements, fewer appointments have been offered [25].

For patients, compliance only makes sense if they quickly derive a benefit from a therapy [26]. If this benefit is not quickly perceptible and comprehensible, the motivation to cooperate decreases [27,28,29].

Therefore, studies suggest that healthcare professionals need to understand reasons for non-compliance if they are to provide supportive care [26].

Elderly patients in particular are recurrently found to be noncompliant with wearing orthoses and following recommended physiotherapy protocols, whether due to cognitive problems with implementation or a lack of therapy slots [16], yet complications and revisions do not seem to occur more frequently in these patients. However, there is a lack of reliable studies concerning this issue. Keeping this in mind, the question arose whether rehabilitation regimens with orthoses and physiotherapy have an influence on the functional outcome and the incidence of complications. It was hypothesised that neither compliance in wearing the orthosis nor adherence to the physiotherapy protocol has an impact on the functional outcome, the incidence of complications, and the revision rate.

## 2. Materials and Methods

The study design is in accordance with the recommendations of the Declaration of Helsinki and was approved by the Ethical Committee of the Medical University of Munich (#22-0019).

### 2.1. Patient Recruitment

Patients with proximal humeral fractures were prospectively identified in the emergency and outpatient department at the Level 1 Trauma Centre at the Munich University Hospital. Patients with a proximal humeral fracture, a minimum age of 18 years, and written informed consent for study participation were included. Exclusion criteria were further injuries of the affected extremity acute and during follow-up, fracture-types AO 11-B3 and 11-C1.3 (which the authors considered as unsuitable for conservative treatment or plate osteosynthesis), open fractures, pathologic fractures resulting from metastatic or primary neoplasia, preoperative non- or malunion of a former fracture, primary bone infections on the effected shoulder, and preoperatively diagnosed neurological deficiency (such as lesions of the axillary or radial nerve).

The participants were separated into four groups: treated with a sling for conservative treatment [A], treated with a sling after locking plate osteosynthesis [B], treated with an abduction brace for conservative treatment [C], and treated with an abduction orthosis after locked plating [D]. The choice of orthosis was determined by the rehabilitation protocols: extraarticular 2-part-fractures (AO 11-A2, AO 11-A3) were immobilised with a sling in both the conservative and operative group. Extraarticular fractures involving the greater tuberosity and bifocal fractures (AO 11-A1, AO 11-B1, AO 11-B2) were immobilised in an abduction orthosis in both the conservative as well as the operative group.

### 2.2. Interventions

Both the doctor on duty and a shoulder specialist (≥50 shoulder surgeries per year) classified all fractures by an X-ray (true anteroposterior and outlet view radiographs), and CT scan. Indications and recommendations for surgical vs. conservative treatment was strictly based on the modified Neer criteria. [30] Regardless of the patient’s choice of treatment, fractures were initially immobilised by the appropriate orthosis depending on the fracture type and study protocol. Furthermore, attention was paid to the organisation and compliance of the physiotherapy.

### 2.3. Rehabilitation Protocols

Rehabilitation was performed according to the inhouse evidence-based guidelines. All patients received their designated orthosis after diagnosis in the emergency room. Non-operatively treated fractures without fracture of the greater tuberosity [A] were recommended to wear a simple sling for 2 weeks. In week 3, they were allowed to start with physiotherapy, such as gentle swinging exercises up to 60° of the flexion and abduction. After 6 weeks, the patients were allowed to lift their arm up to 90° flexion and abduction in the shoulder and the sling could be trained off. From the ninth week onwards, there were no more restrictions, and they could start to regain physical strength. Surgically treated fractures without a fracture of the greater tuberosity [B] could wear the sling in week 1 and 2 for analgetic reasons. They were recommended to start mobilisation exercises with abduction and flexion of a maximum of 60° and an internal/external rotation of 20/0/20° in week 1 and 2. From the third week onwards, they could increase to 80° of flexion and abduction and 30° of rotation (internal and external). After week 4, the patients were allowed to do 120° of abduction and flexion and from the seventh week onwards, they could move their shoulder freely. The patients of group [C] and [D] were treated according to the same protocol. They were recommended to wear the abduction orthosis for 4 weeks. During that time, they were recommended to limit abduction and flexion to 60° with an internal and external rotation of 20°. From week 5 onwards, they were allowed to take off the abduction brace and start with adduction to a neutral position. In addition, they were allowed to perform flexion and abduction up to 80° and an internal/external rotation of 30/0/30°. After week 6, abduction and flexion up to 120° were allowed and then successively increased in a pain-adapted manner.

The detailed rehabilitation protocols can be found in the Appendix A.

### 2.4. Follow Up

Participants were followed up in the outpatient clinic 6 weeks after trauma by interview, clinical examination, and an X-ray. Here, the duration of wearing the orthosis, the number of physiotherapy treatment appointments, and any problems with the orthosis were queried using a questionnaire. If no physiotherapy was conducted, the reason for this was identified. In case of complications or other problems, an earlier additional appointment was offered. This six-week follow-up was defined as the primary end point of the study. Nevertheless, a general overall follow-up according to the hospital standards included reassessments at 3 and 12 months.

### 2.5. Outcome Parameters

The constant score (CS) has been used as the primary outcome parameter to evaluate the functional outcome as described in the original publication by Constant and Murley [31]. This score assigns the subjective parameters of pain, daily function, the objective range of motion, and strength for the glenohumeral joint [32,33]. A maximum of 100 points can be achieved. The muscle strength assessment was performed with a digital force gauge (Mecmesin Ltd., Slinfold, West Sussex, UK). The minimal clinically important difference for the constant score for proximal humerus fractures is calculated at 5.4 and 11.6 points [34]. Furthermore, the following parameters were evaluated in the questionnaire as the secondary outcome parameters: problems with the application of orthosis (yes/no); problems with the orthosis itself (trapping/help needed, increases pain, anxiety, delirium, dyspnoea, bruises, stasis/swelling, constrictions neck, wounds); and quantification of help for putting on the orthosis as well as orthosis wearing period (hours per day). In all cases, fracture healing, secondary dislocation, and implant failure were evaluated by X-ray (true anteroposterior and outlet view radiographs) at the six-week follow-up. The questionnaire can be found in the Appendix A.

### 2.6. Power Calculation and Statistical Evaluation

The primary outcome parameter was the CS (0–100 points). In a case/power calculation for unpaired samples and steady targets, an effect size of 5 points difference at a standard deviation of 10 points was expected. These values were taken from published original articles for both proceedings, checked for plausibility, and implemented. From here, the parameters were as follows: delta = 15, standard deviation 15, alpha = 0.05, and power = 0.8, resulting in n = 63 cases. To safeguard the quality of the study, the dropout rate should not exceed 20% (13 cases). Therefore, a final cohort size of n = 76 was calculated.

Continuous variables were described by the means and standard deviation and were compared using a t-test and one-way ANOVA as well as Tukey’s HSD. Testing for normal distribution was performed using the Shapiro–Wilk test. Categorical variables were analysed using the Chi-square test. The level of significance for all testing was set at *p* < 0.05. Statistical analysis was performed using SPSS (IBM Corp. Released 2016. IBM SPSS Statistics for Windows, Version 24.0. Armonk, NY: IBM Corp.).

## 3. Results

Between August 2019 and January 2022, n = 149 participants were included with a complete follow-up. The mean age of [AB] (n = 62) was 73.9 ± 7.2 years (*p* = 1.2 [A] vs. [B]), and n = 67 (77%) of the participants were female (*p* = 0.08, [A] vs. [B]). The mean age of [CD] (n = 87) was 77.8 ± 6.6 years (*p* = 0.9, [C] vs. [D]), and n = 42 (67.8%) of the participants were female (*p* = 0.07) [C] vs. [D]). The distribution of the fracture pattern groups is shown in Table 1.

In [AB], n = 33 (53.2%) completed orthosis and physiotherapy according to the protocol, n = 10 (16.1%) terminated one or both parameters earlier, and n = 19 (30.6%) terminated both parameters earlier (*p* < 0.001). In [CD], n = 39 (44.8%) completed orthosis and physiotherapy properly, n = 23 (26.4%) finished one or both parameters earlier, and n = 25 (28.7%) terminated both parameters earlier (*p* < 0.001) (Table 2).

The CS in [AB] was 56.8 ± 9.6 among the patients that completed orthosis and physiotherapy properly; it was 64 ± 7.3 in the group that finished one or both parameters earlier and 59.4 ± 10.7 in the group terminating both parameters earlier (*p* = 0.9). The CS in [CD] was 58.2 ± 7.5 among the patients that completed orthosis and physiotherapy properly, 57.5 ± 9.6 in the group that finished one or both parameters earlier, and 60.4 ± 11 in the group terminating both parameters earlier (*p* = 0.7) (Figure 1, Table 3).

In [AB], neither the regular termination of sling therapy (*p* = 0.8) nor the completion of the physiotherapy protocol (*p* = 1.0) influenced the CS significantly. Additionally, no significant differences were found for the regular completion of sling therapy (*p* = 0.1) and the termination of the physiotherapy protocol (*p* = 0.7) in [CD].

In total, n = 55 (36.9%) participants stopped wearing the orthosis earlier than scheduled in the rehabilitation protocol. The reasons of an earlier termination are shown in Figure 2. Overall, the mean length of orthosis wearing was 58.1 ± 36.7% of the time recommended.

A total of n = 66 (44.3%) participants dropped physiotherapy: n = 30 (45.5%) for COVID-19 pandemic reasons, n = 25 (37.9%) for other scheduling issues, and n = 11 (16.7%) no longer considered physiotherapy necessary.

A total of n = 10 patients had a complication, either with varus dislocation of the humeral head or dislocation of the greater tuberosity. The distribution within the evaluated groups is shown in Table 4. Moreover, other complications were detected: n = 3 (2%) patients suffered from resorption of the greater tuberosity, n = 2 (1.3%) cases of avascular necrosis were detected, and n = 1 (0.6%) patient developed pseudoparalysis. In n = 1 (0.6%) case of 45° varus dislocation as well as in n = 1 (0.6%) case of avascular necrosis, an indication for surgical revision was discussed but declined by the participants.

The CS after the 1-year follow-up was completed by n = 131 participants (87.9%). The CS in [AB] after that time was 68.6 ± 9 in the group that completed orthosis and physiotherapy properly; it was 70 ± 4.2 in the group that finished one or both parameters earlier and 75.5 ± 6.8 in the group terminating both parameters earlier (*p* = 0.2). The CS in [CD] was 68.3 ± 8.9 in the group that completed orthosis and physiotherapy according to the rehabilitation protocol; it was 71.2 ± 11 in the group that finished one or both parameters earlier and 72.3 ± 11.8 in the group terminating both parameters earlier (*p* = 0.4).

## 4. Discussion

The aim of the study was to evaluate the influence of a consistent adherence to an orthosis and physiotherapy protocol on the functional outcome after a proximal humeral fracture in the elderly. The fractures were classified by two surgeons and were included in one of the four treatment groups according to the study protocol. In n = 33 (22.1%) cases, previous pathologies such as partial rotator cuff tears, beginning arthrosis, or calcific tendinopathy occurred, which is in accordance with Chard et al. who published a prevalence of 21% shoulder disorders in the elderly [35]. Thus, the studied collective is a representative group. Nevertheless, no patient underwent previous surgery or suffered from acute symptoms. Epidemiological aspects were normally distributed without significant differences in age and gender.

Regarding the sling group, only half of the collective completed the orthosis and physiotherapy protocols as recommended. One third of all participants terminated both parameters earlier. A minority of participants finished only one or both parameters early.

In the abduction brace group, less than half of the patients completed the orthosis and physiotherapy protocols as recommended, barely one third terminated both parameters earlier, one fourth finished one or both early.

This significantly different compliance contrasts a consistently nonsignificant trend in functional outcome. No significant functional advantage was found in either the conservative or operative arm of both orthoses when the orthosis and/or physiotherapy protocol were completed as recommended. Moreover, the multifactorial inability of protocol compliance in this collective showed no negative influence on the complication and revision rates.

Remarkably, when looking more closely at the evaluated parameters, the time patients wore orthoses averaged across all the groups was only 60% of the recommended time according to the protocols. The reasons provided by the patients for the early termination of orthotic treatment need to be evaluated. Mostly, there was not a lack of will, but comprehensible and in part medically alarming reasons. A more intensive introduction of the use of the prescribed orthosis and an explanation of why to wear it, as well as a closer re-evaluation, could be approaches for bringing about an improvement in this regard. Livesey et al. demonstrated that patients who were better educated and thus had a better understanding of the need for orthosis, as well as patients with home support and patients over 60 years of age, were more likely to wear orthoses. However, the mean age in this study was only 59 ± 13.4 years [36]. Compared to the present collective, a relevant difference in compliance, but also in functional demand, must be assumed due to the significantly higher age (73.9 ± 7.2 years) of the participants. The authors hypothesise that with respect to age, there is a cut-off at which the benefits of orthotic compliance have an impact on the outcome.

About two fifths of participants finished the physiotherapy protocol earlier than recommended. The authors assume a lockdown bias due to the COVID-19 pandemic and suspect a lower dropout rate under normal conditions. Regardless, dropout in more than half of all cases under “normal conditions” is also a remarkable value. Burn et al. reported a physiotherapy participating rate of 41% and a significant decrease over time in a collective with a mean age of 45 ± 13 years after rotator cuff repair [37]. Given the vulnerability of this older cohort, it is clear that dramatic deviations from the physiotherapy protocol must occur.

At the one-year follow-up, there was no significant difference in functional outcome between the groups. However, a change at this time was no longer expected, since functionally relevant secondary dislocations appear significantly earlier.

Furthermore, we will describe two individual cases in more detail. A 72-year-old man from group [C] who suffered an AO 11-B2 fracture reported severe discomfort as early as 48 h after receiving an abduction brace. Contrary to the study protocol, a change was made to a simple sling for comfort reasons, which was tolerated for another 48 h before the patient decided to quit immobilisation. Nevertheless, fracture healing was satisfying; the CS after 6 weeks was above the average of 70 points (Figure 3).

An 86-year-old woman with an AO 11-B2 fracture from group [D] required postoperative monitoring on an intermediate care unit (IMC) after an abduction brace was applied in the operating room according to the rehabilitation protocol. After arrival at the IMC, postoperative delirium was diagnosed. In addition to conservative therapy, the patient required a clonidine perfusor for agitation and vegetative symptoms as well as haloperidol for psychotic symptoms. After transfer to the inhouse geriatric trauma centre, the geriatrician examined the patient and removed the brace whereupon the patient recovered soon. By the evening of the same day, she was already without medication for delirium. The patient also reported increasing pain during the aforementioned period in which she wore the orthosis but was unable to articulate it due to delirium.

## 5. Conclusions

A consistent adherence to and compliance with the rehabilitation protocol in regard to orthoses and physiotherapy may be a key factor in achieving good functional outcomes in the treatment of proximal humeral fractures. However, several factors prevent, especially in the elderly, from properly following both. This collective not only has difficulties arranging physiotherapy appointments but also has difficulties arriving to therapy. Furthermore, orthoses can lead to relevant impairments in daily life and, in some cases, even to dangerous situations.

The presented study showed that these, sometimes severe, deviations from the rehabilitation protocols in the elderly do not cause a significant deterioration of the functional outcomes. The authors assume that beyond a certain age, strict adherence to the protocols must be clearly relativised, and that attempting to implement them nevertheless involves more disadvantages and a loss of quality of life for the patients than achieving a relevant improvement in functional outcomes or reducing the complication rate.

## Figures and Tables

**Figure 1 jcm-12-01762-f001:**
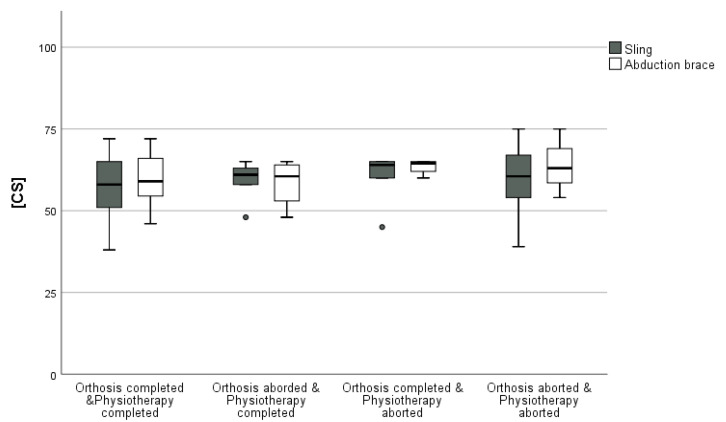
Boxplot of the CS [points] according to compliance collectives.

**Figure 2 jcm-12-01762-f002:**
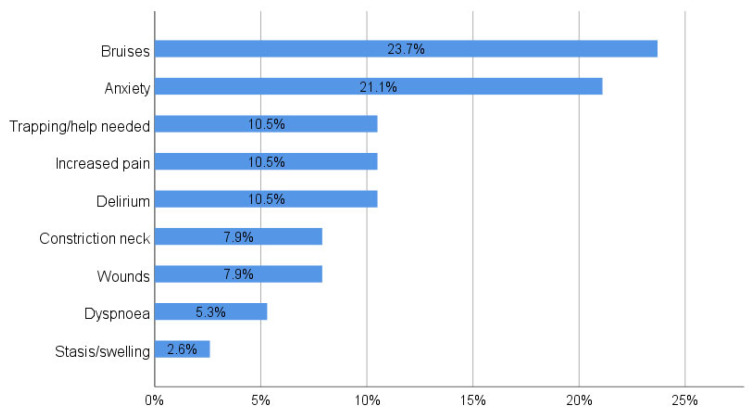
Reasons for early termination of orthosis therapy.

**Figure 3 jcm-12-01762-f003:**
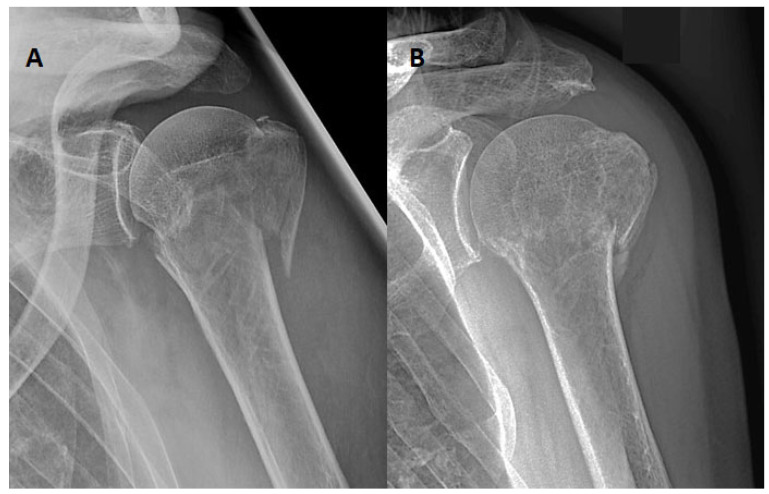
Case of a 72-year-old man of group [C] with an AO 11-B2 fracture, (**A**) anteroposterior X-ray from the fracture day. Patient-related termination of orthosis protocol after 96 h. (**B**) Anteroposterior X-ray after 6 weeks with acceptable healing and CS above the average value of 70 points.

**Table 1 jcm-12-01762-t001:** Distribution of the participants in the four groups according to fracture type.

	AO 11-A	AO 11-B	AO 11-C
[A] Sling conservative	16	1	-
[B] Sling operative	43	1	1
[C] Abduction brace conservative	12	18	1
[D] Abduction brace operative	-	51	5

**Table 2 jcm-12-01762-t002:** Distribution of the number of participants according to compliance collectives.

Orthosis	Physiotherapy	Orthosis Completed (N)	Orthosis Dropout (N)	*p*-Value
Sling (total)	Physiotherapy completed	33 (53.2%)	5 (8%)	<0.001
	Physiotherapy dropout	5 (8%)	19 (30.6%)	
Sling (surgery)	Physiotherapy completed	26 (41.9%)	4 (6.5%)	<0.001
	Physiotherapy dropout	4 (6.5%)	10 (16.1%)	
Sling (conservative)	Physiotherapy completed	7 (11.3%)	1 (1.6%)	0.003
	Physiotherapy dropout	1 (1.6%)	9 (14.5%)	
Abduction brace (total)	Physiotherapy completed	39 (44.8%)	6 (6.9%)	<0.001
	Physiotherapy dropout	17 (19.5%)	25 (28.7%)	
Abduction brace (surgery)	Physiotherapy completed	26 (29.9%)	3 (3.4%)	<0.001
	Physiotherapy dropout	10 (11.4%)	17 (19.5%)	
Abduction brace (conservative)	Physiotherapy completed	13 (14.9%)	3 (3.4%)	0.07
	Physiotherapy dropout	7 (8%)	8 (9.2%)	

**Table 3 jcm-12-01762-t003:** Distribution of the CS according to compliance collectives.

Orthosis	Physiotherapy	Orthosis Completed (N)	Orthosis Dropout (N)	*p*-Value
Sling (total)	Physiotherapy completed	56.8 ± 9.6	66.4 ± 5.9	0.9
	Physiotherapy dropout	61.6 ± 8.7	59.4 ± 10.7	
Sling (surgery)	Physiotherapy completed	59.3 ± 9.7	62.3 ± 5.9	0.4
	Physiotherapy dropout	60.5 ± 7.7	60.5 ± 9.7	
Sling (conservative)	Physiotherapy completed	55.4 ± 9.8	56.7 ± 15	0.6
	Physiotherapy dropout	55.4 ± 10.9	63.3 ± 9.3	
Abduction brace (total)	Physiotherapy completed	58.2 ± 7.5	57.1 ± 8.9	0.7
	Physiotherapy dropout	57.8 ± 10.3	60.4 ± 11	
Abduction brace (surgery)	Physiotherapy completed	53.8 ± 10.1	39	0.9
	Physiotherapy dropout	51	49.3 ± 11	
Abduction brace (conservative)	Physiotherapy completed	48 ± 6.6	45	0.3
	Physiotherapy dropout	61	53.8 ± 11.9	

**Table 4 jcm-12-01762-t004:** Overview of secondary dislocations according to compliance collectives.

Orthosis	Physiotherapy	Orthosis Completed (N)	Orthosis Aborted (N)	*p*-Value
Sling (total)	Physiotherapy completed	5 (6.3%)	1 (1.3%)	1
	Physiotherapy dropout	3 (3.8%)	1 (1.3%)	
Sling (surgery)	Physiotherapy completed	3 (3.8%)	1 (1.3%)	1
	Physiotherapy dropout	3 (3.8%)	1 (1.3%)	
Sling (conservative)	Physiotherapy completed	2 (2.5%)	-	1
	Physiotherapy dropout	-	-	
Abduction brace (total)	Physiotherapy completed	2 (3.8%)	2 (3.8%)	0.4
	Physiotherapy dropout	-	3 (5.8%)	
Abduction brace (surgery)	Physiotherapy completed	1 (1.9%)	2 (3.8%)	1
	Physiotherapy dropout	-	2 (3.8%)	
Abduction brace (conservative)	Physiotherapy completed	1 (1.9%)	-	1
	Physiotherapy dropout	-	1 (1.9%)	

## Data Availability

Date are available at the Department of Orthopaedics and Trauma Surgery, Musculoskeletal University Centre Munich (MUM), University Hospital, LMU Munich, Germany.

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
