# Peer review of "The Influence of Adherence to Orthosis and Physiotherapy Protocol on Functional Outcome after Proximal Humeral Fracture in the Elderly"

_jcm, 2023, doi:10.3390/jcm12051762_

Round 1

Reviewer 1 Report

Clinical management of older adults with proximal humerus fractures is an important topic. However, it is not clear that the design and conduct of the study is sound based on the manuscript as written. The introduction does not provide a strong scientific basis for the research questions. The authors focus on orthosis use and some unclear aspects of physical therapy. No theory is presented to support the specific relationships between these factors and the specific outcomes used in the study. In fact, the authors do not describe how compliance with physical therapy was measured, nor do they describe the physical therapy interventions. Rather, they describe the restrictions on physical therapy. The lack of specificity is a critical flaw in this work. In addition, the methods are not described sufficiently to judge the adequacy of the design or of its execution. Some specific examples and recommendations are provided below. 

Throughout: Provide citations for the evidence to support statements such as “ non-displaced fractures may be treated non-operatively with good functional results,” and “Locking plates have been established for open reduction and internal fixation” what is meant by this statement?  

For the following statements, is there evidence on the use of these approaches?

P 1 line 34-35: “The involvement of the major tubercule plays a decisive role in the choice of orthosis. If it is not affected, the arm is placed in a simple sling. If it is fractured an abduction brace is recommended to neutralise the tension forces of the rotator cuff.”

P 2, lines 38-40: From the section of the introduction on the evidence cited regarding adherence and compliance, it is unclear how well the authors provided a synthesis of the relevant evidence.

Page 2, lines 41-42: “In particular, for the shoulder with its multiple injuries, there are differentiated, evidence-based and elaborated treatment protocols.” This statement implies that there are either clinical practice guidelines or consensus-based recommendations for treatment of the referenced injuries. However, the citations do not appear to refer to such sources.

P 2. Line 43: “For patients compliance only makes sense, when they receive a benefit from a therapy. [11] If it is not quickly perceptible and comprehensible, the motivation to cooperate decreases.”  This statement seems to infer the results of one study to all patients.

Provide evidence to support the main components of the following: P 2, line 45-46 “Elderly patients in particular are recurrently found to be noncompliant with wearing orthoses (REFs) and following recommended physiotherapy protocols,(REFs) whether due to cognitive problems with implementation (REFs) or lack of therapy slots (REFs); yet complications and revisions do not seem to occur more frequently in these patients.(REFs)

The paper begins with the statement that patient compliance is crucial for successful treatment, then go on to present the hypothesis that compliance will not impact outcome. Please clarify

Has a minimal clinically important difference estimate been published for the Constant score in this population?

The authors provide very little detail about the physical therapy protocol. In essence they describe the restrictions, rather than the active physical therapy interventions. At each phase for each group

Outcomes that are reported in the manuscript but the methods used to collect the data are not described. For example, how was the data for the constant score collected? How was determination of completion of orthosis and physiotherapy made and how was the data collected?

Author Response

Dear reviewer,

Thank you very much for the additional comments and suggestions. We have modified the manuscript according to the comments below. We have also updated the literature. Please find here our point-by-point editing: 

Clinical management of older adults with proximal humerus fractures is an important topic. However, it is not clear that the design and conduct of the study is sound based on the manuscript as written. The introduction does not provide a strong scientific basis for the research questions. The authors focus on orthosis use and some unclear aspects of physical therapy.

The entire introduction was revised in view of it, and literature references were added in several sections.

No theory is presented to support the specific relationships between these factors and the specific outcomes used in the study. In fact, the authors do not describe how compliance with physical therapy was measured, nor do they describe the physical therapy interventions. Rather, they describe the restrictions on physical therapy. The lack of specificity is a critical flaw in this work. In addition, the methods are not described sufficiently to judge the adequacy of the design or of its execution. Some specific examples and recommendations are provided below. 

Throughout: Provide citations for the evidence to support statements such as “ non-displaced fractures may be treated non-operatively with good functional results,” and “Locking plates have been established for open reduction and internal fixation” what is meant by this statement?  

The entire manuscript has been revised and additional literature references were included to support statements such as "non-displaced fractures may be treated non-operatively with good functional results." The implementation of locked, anatomically configurated osteosynthesis plates in the early 21st century, along with the implementation of reverse shoulder arthroplasty, was one of the milestones in the surgical treatment of proximal humerus fractures in the last 20 years. The sentence "Locking plates have been established for open reduction and internal fixation" refers to this event. Though, it has been modified to make it easier to understand.

For the following statements, is there evidence on the use of these approaches?

P 1 line 34-35: “The involvement of the major tubercule plays a decisive role in the choice of orthosis. If it is not affected, the arm is placed in a simple sling. If it is fractured an abduction brace is recommended to neutralise the tension forces of the rotator cuff.”

The entire section on the use of orthoses in the rehabilitation of proximal humerus fractures was revised and literature references were included.

P 2, lines 38-40: From the section of the introduction on the evidence cited regarding adherence and compliance, it is unclear how well the authors provided a synthesis of the relevant evidence.

This section was revised, too.

Page 2, lines 41-42: “In particular, for the shoulder with its multiple injuries, there are differentiated, evidence-based and elaborated treatment protocols.” This statement implies that there are either clinical practice guidelines or consensus-based recommendations for treatment of the referenced injuries. However, the citations do not appear to refer to such sources.

A reference to the 2021 ESTES recommendations for the treatment of proximal humerus fractures in elderly patients was included.

P 2. Line 43: “For patients compliance only makes sense, when they receive a benefit from a therapy. [11] If it is not quickly perceptible and comprehensible, the motivation to cooperate decreases.”  This statement seems to infer the results of one study to all patients.

            Further literature has been added.

Provide evidence to support the main components of the following: P 2, line 45-46 “Elderly patients in particular are recurrently found to be noncompliant with wearing orthoses (REFs) and following recommended physiotherapy protocols,(REFs) whether due to cognitive problems with implementation (REFs) or lack of therapy slots (REFs); yet complications and revisions do not seem to occur more frequently in these patients.(REFs)

Unfortunately, there is hardly any literature on this topic. A sentence referring to to this issue, pointing out the lack of valid studies has been added.

The paper begins with the statement that patient compliance is crucial for successful treatment, then go on to present the hypothesis that compliance will not impact outcome. Please clarify

            The section was revised.

Has a minimal clinically important difference estimate been published for the Constant score in this population?

The minimal clinically important difference for the Constant score for proximal huemrus fractures was included to the Materials & Methods section.

The authors provide very little detail about the physical therapy protocol. In essence they describe the restrictions, rather than the active physical therapy interventions. At each phase for each group

The section of the rehabilitation protocols was revised. Furthermore, the protocols have been added to the supplementary data.

Outcomes that are reported in the manuscript but the methods used to collect the data are not described. For example, how was the data for the constant score collected? How was determination of completion of orthosis and physiotherapy made and how was the data collected?

The constant score is a common score for assessing everyday shoulder function and complaints. The maximum number of points that can be achieved is 100, and it is measured using predefined questionnaires published in the mentioned publications. Among others, mobility, everyday functions and pain are queried and the muscle strength of the affected arm is measured. These informations as well as the name of the force gauge were added tot he section. Furthermore, the questionnaire used was translated and added tot he supplementary data.

We hope this is sufficient for the manuscript to be accepted for publication.

On behalf of all the co-authors Yours sincerely

Tobias Helfen

Reviewer 2 Report

Although it is generally believed that in the treatment of proximal humeral fractures, continuous adherence and physical therapy may be the key factors to achieve good functional results. However, in daily life, many patients can not adhere to the above treatment plan because of various factors. This article concluded by retrospective follow-up study that although they can not adhere to the strict physical treatment plan, there is no significant difference in the final clinical results. This conclusion has certain clinical guiding significance and is also interesting. The author determines the surgical and physical treatment plan according to the AO classification of proximal humeral fracture, however  the AO classification of type A, B and C is also divided into many subtypes, involving the fracture and displacement of relatively complex greater and lesser tuberosity. Generally, it is not necessary to use sling and brace for external fixation when open reduction is used to obtain strong internal fixation.  However, the author chooses whether to use sling and brace for external fixation only according to the type of fracture, think it is inappropriat .

Author Response

Dear reviewer,

Thank you very much for the additional comments and suggestions. We have modified the manuscript according to the comments below. We have also updated the literature. Please find here our point-by-point editing:

Although it is generally believed that in the treatment of proximal humeral fractures, continuous adherence and physical therapy may be the key factors to achieve good functional results. However, in daily life, many patients can not adhere to the above treatment plan because of various factors. This article concluded by retrospective follow-up study that although they can not adhere to the strict physical treatment plan, there is no significant difference in the final clinical results. This conclusion has certain clinical guiding significance and is also interesting. The author determines the surgical and physical treatment plan according to the AO classification of proximal humeral fracture, however  the AO classification of type A, B and C is also divided into many subtypes, involving the fracture and displacement of relatively complex greater and lesser tuberosity. Generally, it is not necessary to use sling and brace for external fixation when open reduction is used to obtain strong internal fixation.  However, the author chooses whether to use sling and brace for external fixation only according to the type of fracture, I think it is inappropriat .

The manuscript, especially the introduction and the material & methods section were revised. In particular, literature references were added to show the scientific background on which the authors' assumptions are based.      

We hope this is sufficient for the manuscript to be accepted for publication.

On behalf of all the co-authors Yours sincerely

Tobias Helfen

Round 2

Reviewer 1 Report

The changes substantially improved the quality of this manuscript.